# TrkB Inhibits the BMP Signaling-Mediated Growth Inhibition of Cancer Cells

**DOI:** 10.3390/cancers12082095

**Published:** 2020-07-28

**Authors:** Min Soo Kim, Wook Jin

**Affiliations:** Laboratory of Molecular Disease and Cell Regulation, Department of Biochemistry, School of Medicine, Gachon University, Incheon 21999, Korea; kimms@gachon.ac.kr

**Keywords:** TrkB, bone morphogenetic proteins (BMPs), BMP type I (BMPRI) and BMP type II (BMPRII) receptor, tumor suppressor activity, tumor progression

## Abstract

We have previously observed that tropomyosin receptor kinase B (TrkB) induces breast cancer metastasis by activating both the Janus kinase 2/signal transducer and activator of transcription 3 (JAK2/STAT3) and phosphatidylinositol-3-Kinase (PI3K)/AKT signaling pathways and inhibiting runt-related transcription factor 3 (RUNX3) and kelch-like ECH-associated protein 1 (KEAP1). These studies indicated that TrkB expression is crucial to the pathogenesis of breast cancer. However, how TrkB regulates bone morphogenetic protein (BMP) signaling and tumor suppression is largely unknown. Herein, we report that TrkB is a key regulator of BMP-mediated tumor suppression. TrkB enhances the metastatic potential of cancer cells by promoting cell anchorage-independent growth, migration, and suppressing BMP-2-mediated growth inhibition. TrkB inhibits the BMP-mediated activation of SMAD family member 1 (SMAD1) by promoting the formation of the TrkB/BMP type II receptor complex and suppresses RUNX3 by depleting BMP receptor I (BMPRI) expression. In addition, the knockdown of TrkB restored the tumor-inhibitory effect of BMP-2 via the activation of SMAD1. Moreover, the TrkB kinase activity was required for its effect on BMP signaling. Our study identified a unique role of TrkB in the regulation of BMP-mediated growth inhibition and BMP-2-induced RUNX3 expression.

## 1. Introduction

The bone morphogenetic protein (BMP) family, a member of the transforming growth factor-β (TGF-β) superfamily, consists of over 20 multifunctional growth factors with a broad range of regulatory functions, including embryonic development, apoptosis, chemotaxis, proliferation, and differentiation [1,2,3,4]. As part of the TGF-β signaling pathway, BMP-2 is secreted when the BMP receptor type I (BMPRIA) and II (BMPRII) complexes on the cell surface, which leads to the activation, and therefore nuclear translocation of SMAD family member 1 (SMAD1), in association with SMAD4, to regulate specific target gene expression [5,6,7].

Similar to the functions of TGF-β in the cancer microenvironments, BMPs, and their receptors play a dual tumor-suppressive and -promoting role in cancer. For example, in an analysis of large patient cohorts, Gremlin 1, a BMP-antagonist, was found to be significantly increased in the tumor tissues of breast cancer patients [8]. In another study, Gremlin 1 was found to be responsible for the tumorigenic potential of cancer cells by inhibiting BMP/SMAD signaling, and its expression was associated with poor prognosis in breast cancer patients [9]. Moreover, estrogen treatment in these cells further reduced BMPRI expression [10], which, along with BMPRII, was also found to be significantly reduced in patients with poorly differentiated prostate, renal, and bladder cancers [5,11,12]. BMPRI expression is frequently reduced and associated with poor survival in pancreatic cancer patients, and the inactivation of BMP signaling promotes tumor invasion [6]. BMP-2 has been found to induce glioma apoptosis by activating BMPRI [13]. However, other studies suggest that BMP signaling promotes tumorigenesis and metastasis. For instance, inhibition of BMP signaling by BMPRI antagonists (alk2, 3, and 6) decreased the growth of lung cancer cells, eventually leading to their death [14]. In addition, the loss of BMPRIB has been associated with poor survival in breast cancer patients and drug resistance [15]. Therefore, the role that BMP signaling plays in tumor progression is still not completely clear.

Brain-derived neurotrophic factor (BDNF), a neurotrophin, interacts with and activates tropomyosin receptor type B (TrkB) and plays a key role in the survival, differentiation, and maintenance of neurons [16]. Recent studies, including ours, have demonstrated that BDNF and TrkB play a crucial role in tumorigenesis and cancer metastasis, and are associated with poor survival in patients with various cancer types. BDNF/TrkB signaling is activated in several cancers, including breast [17], colon [18], lung [19], pancreatic [20], and ovarian cancers [21], cutaneous melanoma [22], and oral squamous cell carcinoma (OSCC). Additionally, its expression is associated with poor prognosis in melanoma and OSCC patients [23]. TrkB inhibition using the small molecule inhibitor ANA-12 hindered medulloblastoma cell survival by inducing apoptosis and reducing extracellular-regulated kinase (ERK) activity [24]. We have previously demonstrated that TrkB induces breast cancer metastasis by activating the Janus kinase 2/signal transducer and activator of transcription 3 (JAK2/STAT3) and phosphatidylinositol-3-Kinase (PI3K)/AKT signaling pathways [25], and suppressing runt-related transcription factor 3 (RUNX3) and kelch-like ECH-associated protein 1 (KEAP1) expression [17]. A previous report linked BMP signaling with RUNX3 (a tumor suppressor) in colorectal carcinoma, where the BMP-2 and BMP-4-mediated upregulation of RUNX3 expression inhibited tumor progression [26]. Based on these studies, we hypothesized that TrkB expression might regulate BMP-mediated tumor suppression to enhance tumorigenicity and cancer metastasis. However, the mechanisms by which TrkB promotes these characteristics via BMP and RUNX3 and the signaling mechanisms that modulate the tumor-suppressive activities of BMP are unclear. Therefore, we have focused on the mechanistic link between TrkB and BMP signaling in the regulation of tumor invasion and tumorigenesis.

## 2. Results

### 2.1. TrkB Expression Inhibits BMP Signaling Cascade

To understand the correlation between TrkB and BMP signaling, we examined whether TrkB expression can modulate the transcription activity of a luciferase gene driven by the BMP response element (BRE). RIE-1, HeLa, and NMuMG cells were transiently-transfected with TrkB, which reduced BRE-luciferase activity relative to that of the control (Figure 1A,B and Appendix A). Thus, TrkB could promote tumor growth via the inhibition of BMP signaling. To test this hypothesis, we generated RIE-1 and HeLa cells that overexpressed TrkB (data not shown).

As expected, the BMP-2-mediated transcriptional activity was reduced by TrkB overexpression in both TrkB-overexpressing cell lines (Figure 1C and Appendix A). In normal RIE-1 and HeLa cells, the BMP-2-mediated phosphorylation of SMAD1 was readily detectable, but this was markedly reduced in the TrkB-overexpressing cells (Figure 1D,E). To further understand the function of TrkB in BMP signaling, we investigated whether TrkB attenuates the growth inhibitory effect of BMP-2. BMP-2 induced growth inhibition in RIE-1 cells but was ineffective against RIE-1-TrkB cells (Figure 1F). These results suggest that TrkB may modulate BMP-2 growth inhibition.

### 2.2. The Loss of TrkB Restores BMP-Mediated Tumor-Suppressive Activities

To further substantiate the role that BMP signaling plays in regulating tumor invasion and the function of TrkB, we knocked down TrkB expression in MDA-MB-231 and Hs578T cells, which have high TrkB expression (Appendix A). We found that TrkB knockdown significantly increased BMP-2-associated BRE transcriptional activity (Figure 2A,B). We also observed that BMP-2 stimulated SMAD1 phosphorylation in MDA-MD-231 TrkB-shRNA cells, but not in control-shRNA cells (Figure 2C). We then examined whether TrkB knockdown restored the inhibitory effect of BMP-2 on growth and found that while MDA-MB-231 control-shRNA cells were resistant to BMP-2, TrkB knockdown cells responded to BMP-2 by attenuating growth (Figure 2D). We also investigated whether TrkB regulates the expression of the BMP antagonist Gremlin 1. Interestingly, the knockdown of TrkB markedly reduced Gremlin 1 expression. In addition, Gremlin 1 expression was markedly reduced by BMP-2 treatment (Appendix A). These results indicate that TrkB plays a role in suppressing the growth inhibitory effect of BMP-2.

### 2.3. TrkB Directly Interacts with BMP Type II Receptors to Inhibit BMP Signaling

Our results suggest that TrkB suppresses BMP-mediated tumor inhibition by regulating the processes upstream of SMAD1. We speculate that the mechanism underlying this process might be associated with TrkB-BMP receptor interactions. We found that TrkB directly interacted with BMPRII, but not with BMPRI (Figure 3A,B). In addition, TrkB and BMPRII were colocalized in the cytoplasm (Appendix A). We then examined the endogenous interaction between TrkB and BMPRII in MDA-MB-231 cells, which expressed TrkB, and the tissues of breast cancer patients. Endogenous TrkB directly interacted with BMPRII, and, in breast cancer patients, TrkB was upregulated in tumor tissues versus healthy tissues. The interaction between endogenous TrkB and BMPRII in these patient tissues was also confirmed by immunoprecipitation (Figure 3C,D).

Moreover, when the cytoplasmic and N-terminal regions of BMPRII, which included the extracellular and transmembrane domains, were deleted, the binding affinity for TrkB remained unchanged. However, when an area between amino acids 203 and 350, which corresponded to the tyrosine kinase domain of BMPRII, was removed, this abolished its binding affinity (Figure 3E). Thus, the kinase domain of BMPRII is crucial for its interaction with TrkB.

### 2.4. The Tyrosine Kinase Activity of TrkB is Required for the Inhibition of BMP-2 Signaling

We assessed the effects of a TrkB kinase inhibitor on BMP-mediated tumor inhibition in RIE-1 and RIE-1-TrkB cells. The inhibition of TrkB kinase activity with K252a significantly increased BRE transcriptional activity in RIE-1-TrkB cells upon treatment with BMP-2, whereas no change to BRE transcription was observed in RIE-1 cells, with or without K252a treatment (Figure 4A). Similar results were obtained using a thymidine incorporation assay. The ability of TrkB to reverse BMP-mediated effects was significantly enhanced in RIE-1-TrkB cells, whereas RIE-1-TrkB cells treated with K252a reversed the effect of TrkB on BMP (Figure 4B).

Similarly, in MDA-MB-231 and Hs578T breast cancer cells, which expressed endogenous TrkB, the BRE transcriptional activity increased in response to K252a and BMP-2 treatments (Figure 4C,D). Consistent with its effect on BRE-luciferase activity and growth, K252a significantly increased the level of SMAD1 phosphorylation in BMP-2-treated RIE-1-TrkB, MDA-MB-231, and Hs578T cells, but not in BMP-2-treated control cells (Figure 4E and Appendix A). Additionally, we were intrigued by the cell-specific effects of K252a on BMP-2 signaling because it implies that TrkB activation, via its overexpression, may be required for K252a to suppress BMP signaling. Therefore, we first validated the notion that TrkB is activated by its overexpression using an immunoprecipitation experiment. Our results showed that in comparison to RIE-1 cells, RIE-1-TrkB cells exhibited a significant increase in TrkB phosphorylation, regardless of whether these cells were treated with BDNF (Appendix A). These results indicate that, upon activation, TrkB reverses BMP-induced growth inhibition.

To assess the effects of TrkB activation on BMP-associated transcriptional activity, we generated RIE-1 TrkB kinase-dead (KD) cell lines using vectors that expressed a kinase-dead mutant of TrkB (K588M). As expected, both RIE-1-TrkB and HeLa-TrkB cells exhibited reduced reporter activity in response to BMP-2 treatment, whereas in cells that expressed the TrkB KD, BMP-2 treatment increased reporter activity (Figure 5A,B). In addition, although TrkB expression inhibited SMAD1 phosphorylation, TrkB K588M failed to do so (Figure 5C).

We were also interested in the possibility that TrkB KD and BMPRII form a complex. Wild-type TrkB was strongly associated with BMPRII; however, the interaction between BMPRII and the TrkB KD mutant was significantly reduced with or without BMP-2 treatment. Furthermore, BMPRII phosphorylation following BMP-2 treatment was observed only in TrkB wild-type cells (Figure 5D). Because BMP-2 stimulates the interaction between BMPRI and BMPRII, we speculate that the formation of a BMPRII-TrkB complex may hinder this interaction. Our results show that BMPRII is strongly bound to BMPRI following treatment with BMP-2, but this interaction was completely abolished in the presence of TrkB (Figure 5E), but not by the TrkB KD mutant (Figure 5F), indicating that TrkB kinase activity is required for it to interfere with BMP-mediated tumor suppression.

### 2.5. TrkB Regulates BMP Type I Receptor Expression to Inhibit BMP Signaling

BMP-2 induces the expression of the tumor suppressor RUNX3 and consequently decreases c-Myc expression when RUNX3 binds to the c-Myc promoter. Moreover, in comparison to the constitutively-active BMPRIA (CA-ALK3), the transfection of a dominant-negative version of BMPRI (DN-ALK3), in the presence of BMPRII, does not suppress c-Myc transcription activity [26], indicating that the activation of BMP signaling by the phosphorylation or overexpression of BMPRI might be required for RUNX3-mediated c-Myc inhibition. We have previously shown that TrkB promotes tumorigenesis and metastasis in breast cancer via the suppression of RUNX3 expression [17]. Our present study shows that TrkB does not interact with BMPRI (Figure 3A). However, it is still not completely clear how TrkB suppresses RUNX3 expression via the inhibition of BMP signaling. We speculate that TrkB might attenuate the BMP inhibition of tumor growth by regulating BMPRI expression and thereby deplete the RUNX3 protein. To verify this hypothesis, we examined the expression of BMPRI and BMPRII. The expression of BMPRI mRNA and protein was significantly reduced in HeLa-TrkB and RIE-1-TrkB cells relative to their wild-type counterpart (Figure 6A and Appendix A) but did not affect BMPRII expression. We next inhibited TrkB kinase activity by introducing a TrkB KD mutant into RIE-1-TrkB cells, which restored BMPRI expression but also did not affect BMPRII expression (Figure 6B and Appendix A). Moreover, the knockdown of TrkB in MDA-MB-231 and Hs578T cells induced BMPRI (Figure 6C–F and Appendix A) but not BMPRII (Appendix A) expression. Moreover, we examined BMPRI expression in the lungs of mice injected with MDA-MB-231 control-shRNA or TrkB-shRNA cells. In comparison to the mice bearing MDA-MB-231 control shRNA cells, BMPRI expression in the lungs of MDA-MB-231 TrkB-shRNA-bearing mice was significantly reduced (Figure 6G). However, BMPRII expression did not change (Appendix A).

Our results show that BMPRI-BMPRII complex formation was significantly suppressed in the presence of TrkB (Figure 5E,F). Under these conditions, we examined whether BMPRI induces the RUNX3 expression in response to BMP-2. RUNX3 expression was markedly upregulated following the introduction of BMPRI in RIE-TrkB cells, which was enhanced by the addition of BMP-2 (Figure 6H); however, RUNX3 expression was not induced by the transfection of the BMPRII gene (Appendix A). Moreover, BMP-2 treatment or BMPRI transfection induced RUNX3 expression in MDA-MB-231-TrkB shRNA cells, which was enhanced when both BMP-2 and BMPRI were applied (Figure 6I). However, RUNX3 expression did not change upon transfection with BMPRII (Appendix A). These results suggest that TrkB inhibits RUNX3 expression by depleting BMPRI expression.

### 2.6. TrkB Promotes Invasion Ability of RIE-1 Cells

Our results suggest that TrkB expression might enhance the tumorigenic potential of the tumor by countering BMP-2-mediated growth inhibition. Based on the above findings, we evaluated whether TrkB overexpression inhibited RIE-1 cell metastasis. To metastasize to distant organs, cancer cells must overcome anoikis as part of cancer progression [27]. We first examined whether TrkB affects anoikis and thereby exhibits the hallmarks of metastasis: invasion and dissemination. In comparison to RIE-1 cells, RIE-1-TrkB cells formed large aggregates in suspension, which reflects anchorage-independent growth (Figure 7A). In the cell migration assay, RIE-1-TrkB cells exhibited markedly enhanced migration ability in comparison to RIE-1 cells, which indicated that the migrative capacity of the cell was correlated with the level of TrkB expression (Figure 7B). We then investigated whether TrkB affected the ability of RIE-1 cells to form colonies and found that RIE-1-TrkB cells formed 3 to 4-fold more colonies than RIE-1 cells (Figure 7C). Next, we performed an in vitro mammosphere formation assay, which measures mammary stem-like traits [28]. Relative to the parental RIE-1 cells, RIE-1-TrkB cells were 2.6-fold more likely to form mammospheres (Figure 7D). Besides, using a wound-healing assay, we showed that TrkB overexpression in RIE-1 cells significantly increased both cell motility and growth beyond that of their parental counterpart (Figure 7E and Appendix A), indicating that TrkB enhances tumor outgrowth. These findings also suggest that TrkB promotes the invasion ability of the tumor cells.

## 3. Discussion

We have previously shown that TrkB induces tumorigenesis and metastasis via the induction of the JAK2/STAT3 and PI3K/AKT pathways. In this study, we showed that TrkB modulates BMP-mediated the growth inhibition of cancer cells. BMP-2 treatment induced SMAD1 phosphorylation and downstream transcription in RIE-1 and HeLa cells, which were significantly reduced by TrkB. In contrast, the loss of TrkB in highly metastatic breast cancer cell lines (MDA-MB-231 and Hs578T) restored BMP signaling. Moreover, growth inhibition induced by BMP-2 treatment was rescued when TrkB was also knocked down in these cells. Our results indicate that the TrkB-mediated suppression of BMP activities is necessary for the induction of tumorigenicity and cancer metastasis, which also supports results from our previous study. We have previously demonstrated that TrkB expression is dramatically increased in triple-negative breast cancers (TNBCs) relative to other subtypes and induces the growth and metastasis of breast cancer cells in vivo [17,25]. In addition, BDNF/TrkB plays a critical role in promoting the recurrence of breast cancer after chemotherapy, cancer stem cell (CSC)-associated tumor-initiating potential, and chemoresistance [27]. Recurrent TNBCs isolated from patients after chemotherapy exhibited an increase in BDNF expression, which promoted CSC self-renewal by increasing aldehyde dehydrogenase 1 (ALDH1) expression. TrkB+ CSCs exclusively overlapped with the ALDH1+ population and treating TrkB+/ALDH1+ cells with BDNF induced Krüppel-like factor (KLF4; a stem cell marker) expression.

Furthermore, when the TrkB+ CSCs were eradicated, TNBC recurrence was prevented, which increased patient survival. In our study, TrkB overexpression in RIE-1 cells markedly increased anchorage-independent growth, migration, and mammosphere formation. These results suggest that TrkB may regulate cancer progression and recurrence by acquiring CSC traits.

We also investigated whether TrkB can regulate the expression of BMP antagonists to inhibit BMP signaling. We found that TrkB significantly upregulated the expression of Gremlin 1 (a BMP antagonist). A previous study demonstrated that BMP signaling prevents TGF-β-mediated CSC transition in breast cancer and TGF-β-induced epithelial-to-mesenchymal transition (EMT) in renal fibrosis. In a mesenchymal subpopulation (MSP) of the human breast mammary epithelia, BMP antagonists (Chordin-like 2 and Gremlin 1) were significantly increased, whereas BMP ligands (BMP-2 and BMP-4) were markedly decreased. Additionally, BMP-4 treatment inhibited the capacity of HMLE, an MSP that had arisen from immortalized human mammary epithelial cells (MECs), to migrate and form mammospheres. The twist is overexpressed in HMLE cells [28,29]. Moreover, Gremlin 1 was identified as a promoter of stem cell maintenance in glioma and colorectal cancer [30,31], and the upregulation of Gremlin 1 significantly correlated with stem cell marker expression and poor survival in estrogen receptor (ER)-negative breast cancer patients [8]. These observations, from ourselves and other researchers, support the hypothesis that TrkB plays a key role in enhancing CSC transition in cancer cells.

Our studies (both past and present) showed that TrkB represses RUNX3 expression by activating the PI3K/AKT signaling pathway and inhibiting the canonical BMP signaling. Introducing BMPRI in the presence of TrkB upregulated RUNX3 expression, which is further enhanced by the addition of BMP-2. Furthermore, upon transfection with BMPRI, RUNX3 expression is increased in MDA-MB-231 TrkB-shRNA but not in MDA-MB-231 control cells, and this increase is further enhanced by BMP-2 treatment. The activity of RUNX3, a tumor suppressor, is inhibited in various cancers by mislocalization, hypermethylation, or loss of heterozygosity [32,33,34]. RUNX3 complexes with and suppresses the function of the Yes-associated protein (YAP), which mediates tumorigenicity and metastatic potential [35,36,37,38].

Additionally, BMP signaling plays a role in the Hippo signaling pathway by inducing the phosphorylation of YAP (inactivated form) and stimulating the Ras association domain family (RASSF1; an upstream regulator of the Hippo signaling pathway) to inhibit tumor development and progression. BMP-2 treatment activates the Hippo kinase cascade, which includes mammalian Ste20-like kinase 1 (MST1), MOB kinase activator 1 (MOB1), and YAP (inactivation), and this action is tumor suppressive. BMP-2 also induces BMPRII expression, apoptosis by inhibiting the nuclear translocation of YAP, and RASSF1-MST1 complex formation by disrupting AKT-RASSF1 interaction [39]. Another report showed that canonical BMP signaling induces the expression of RUNX3, which suppresses tumor growth by binding to the RUNX-binding elements on the c-Myc promoter and inhibiting its transcription [26]. Our results provide further evidence for a link between BMP signaling and RUNX3.

To date, the link between TrkB and BMP signaling is unknown, and our study is the first to report this association. We have identified a unique role for TrkB in the regulation of BMP and RUNX3-mediated growth inhibition of cancer cells.

## 4. Materials and Methods

### 4.1. Cell Lines, Culture Conditions and Chemical Inhibitors

RIE-1, HeLa, NMuMG, 293T, and human highly metastatic cancer cells (Hs578T and MDA-MB-231) cells were cultured in Dulbecco’s modified Eagle’s medium (DMEM) supplemented with 10% fetal bovine serum (FBS). BMP-2 from PeproTech (500-P195BT) used at the final concentration of 50 ng/mL for the indicated time. The protein kinase inhibitor K252a was purchased from Abcam.

### 4.2. Plasmids and Viral Production

RIE-1, HeLa, and NMuMG cells were infected with a V5-TrkB using a pLenti6.3/V5-TOPO TA Cloning Kit (Invitrogen, Waltham, MA, USA) to generate TrkB overexpression cells [25] and the TrkB K588M mutant using previously described [40] was generated by site-directed mutagenesis with a Site-Directed Mutagenesis Kit (ThermoFisher Scientific, Lafayette, CO, USA). To generate a stable knockdown of TrkB, small hairpin-expressing vectors were purchased from Sigma-Aldrich (SHCLNG-NM_006180). Production and infection of target cells were previously described [25] and selected with 2 µg/mL puromycin and 500 µg/mL G418. Plasmid transfections were carried out using Lipofectamine 2000 (Invitrogen, Waltham, MA, USA) reagent according to the manufacturer’s instructions.

### 4.3. Human Breast Tumor Samples

Proteins extracted from human breast normal and tumor samples obtained as previously described [17,41].

### 4.4. Antibodies, Western Blotting, Immunoprecipitation, and Immunofluorescence

We performed Western blotting, immunoprecipitation, and immunofluorescence analysis, as previously described [42]. phospho-SMAD1 (ab214423), SMAD1 (ab63356), BMPRI, BMPRII, phospho-tyrosine, phosphor-Ser/Thr, and TrkB were from Abcam. Flag and β-actin were from Sigma-Aldrich; V5 was from Invitrogen; HA, Myc, and GFP were from Santa Cruz Biotechnology. DyLight594, DyLight488, and DAPI were from VECTOR.

### 4.5. Invasion, Anoikis, Wound Healing, Anchorage-Independent Cell Growth, and Mammosphere Assays

All the assays were performed as previously described [25,43,44]. For anchorage-independent cell growth and soft agar assays, 1 × 10^3^ cells/well seeded into 6-well cell culture plates. For the wound healing assay, 1 × 10^6^ cells/well seeded into 6-well cell culture plates. For the invasion assay, 1 × 10^4^ cells seeded into 24-well BD Matrigel invasion chambers with 8-µm pores (Corning, Salt Lake City, UT, USA.62405-744). For anoikis assay, RIE-1 and RIE-1-TrkB cells were seeded into an Ultra-Low Cluster plate (Corning, Salt Lake City, UT, USA) at 1 × 10^5^ cells per well in a six-well plate and photographed at 7 days. For mammosphere assay, 1 × 10^3^ cells/well were seeded into 96-well ultra-low adhesion plates in DMEM medium.

### 4.6. Luciferase Reporter Assay

3 × 10^4^ cells were transfected with a BRE-Luciferase reporter plasmid using Lipofectamine 2000 (Invitrogen, Waltham, MA, USA). The cell lysates collected 48 hr after transfection, and the luciferase activities were measured using the Enhanced Luciferase Assay Kit (BD Biosciences, San Jose, CA, USA).

### 4.7. RNA Preparation and RT-PCR Analysis

Total RNA was isolated using RNeasy Mini Kits (Qiagen, Frederick, MD, USA), and RT-PCR analysis was performed using a One-Step RT-PCR kit (Qiagen, Frederick, MD, USA) according to the manufacturer’s instructions.

For quantitative RT-PCR, reverse transcription performed with the Superscript Ⅳ First-strand synthesis system (Invitrogen Waltham, MA, USA) and amplification performed with SYBR Green Mix I (Roche, Pleasanton, CA, USA), and All PCR analyses were conducted in triplicate using the 7900HT Fast Real-Time PCR System (Applied Biosystems, Beverly, MA, USA). The primer sequences used to amplify the investigated genes listed in Appendix A.

### 4.8. Cell Proliferation Assay

Actively growing asynchronous cells were plated in 24-well plates at a density of 0.5 × 10^5^ cells/well in 0.5 mL culture medium. After waiting 4 hr for surface attachment, we incubated the cells in the presence or absence of porcine BMP-2 (5 ng/mL) for 24 hr. The cells were then pulse-labeled with 0.5 µCi [3H]-thymidine for 2 hr, fixed with 1 mL methanol/acetic acid, 3:1 (vol/vol), for 1 hr at 25 °C, washed twice with 2 mL 80% methanol, incubated at 37 °C with 0.2 mg/mL trypsin for 30 min and then solubilized with 0.5 mL 1% sodium dodecyl sulfate (SDS). Incorporated activity levels were measured with a scintillation counter.

### 4.9. Statistical Analysis

Data expressed as the means ± SEM. Statistical analyses of the data conducted via the Student’s *t*-test (two-tailed). Differences were considered statistically significant at *p* < 0.001.

## 5. Conclusions

TrkB promotes the formation of TrkB/BMPRII complex by interacting between the TrkB kinase domain and then TrkB suppresses activation of SMAD1. Eventually, TrkB suppresses BMP and RUNX3-mediated growth inhibition of cancer cells.

## Figures and Tables

**Figure 1 cancers-12-02095-f001:**
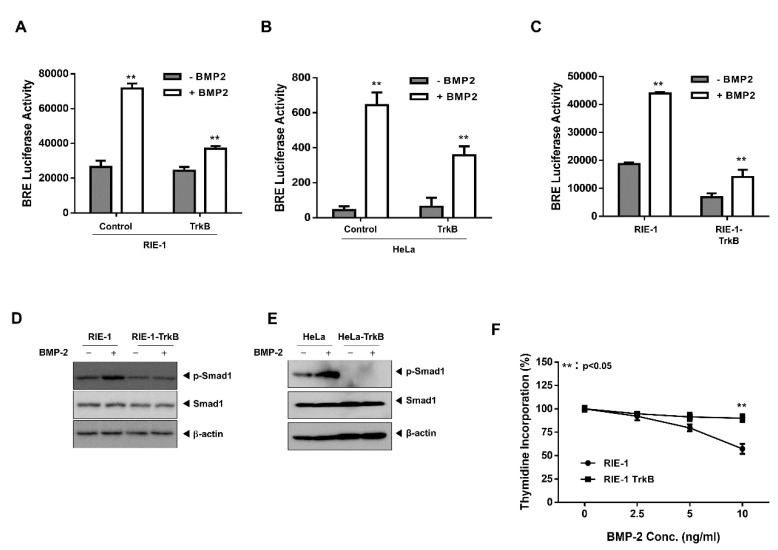
Tropomyosin receptor type B (TrkB) overexpression inhibited the tumor-suppressive activities of bone morphogenetic protein (BMP)-2. (**A**,**B**) BMP-2-responsive BMP response element (BRE) Luciferase reporter activity in rat intestinal epithelial cells (RIE-1). (**A**) and HeLa cells (**B**) transfected with the TrkB construct. Luciferase activity was measured 24 h after treatment with BMP-2 (5 ng/mL). ** Control versus treatment with BMP-2, *p* < 0.05, *n* = 3. (**C**) BMP-2-responsive BRE Luciferase reporter activity in RIE-1 and RIE-1-TrkB cells. ** Control versus treatment with BMP-2, *p* < 0.05, *n* = 3. (**D**,**E**) Western blot analysis of phospho- SMAD family member 1 (SMAD1) and SMAD1 expression in RIE-1 or RIE-1-TrkB cells (**D**), or in HeLa or HeLa-TrkB cells (**E**) after stimulation with BMP-2 (5 ng/mL). (**F**) Thymidine incorporation assay of RIE-1 and RIE-1-TrkB cell proliferation following treatment with various BMP-2 concentrations. The points represent the means from three measurements ± SD. ** RIE-1 versus RIE-1-TrkB, *p* < 0.05, *n* = 3. * RIE-1 versus RIE-1-TrkB, *p* < 0.03, *n* = 3.

**Figure 2 cancers-12-02095-f002:**
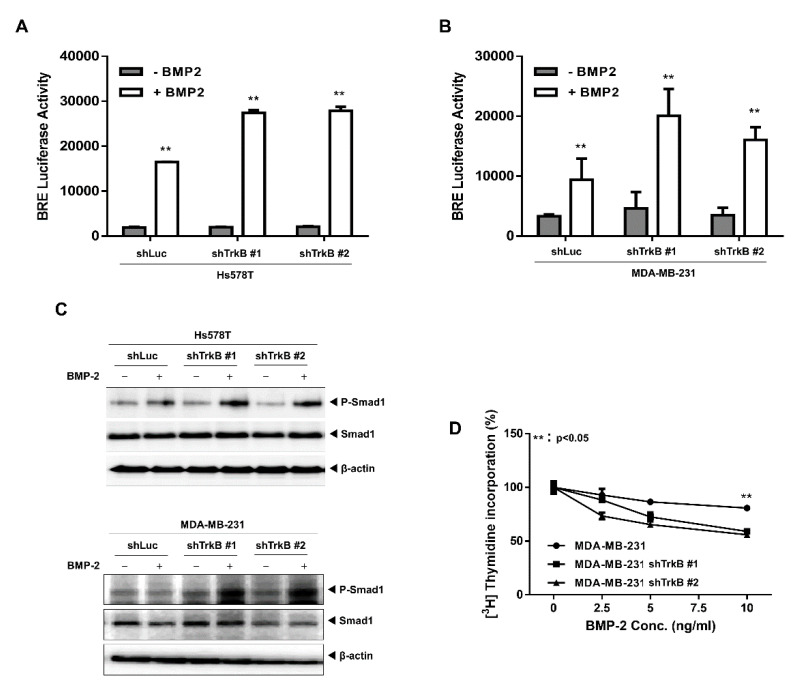
Loss of TrkB in highly metastatic breast cancer cells inhibited BMP signaling. (**A**) BMP-2-responsive BRE Luciferase reporter activity in Hs578T cells transfected with the control or TrkB short hairpin RNA (shRNA). Luciferase activity was measured 24 h after treatment with BMP-2 (5 ng/mL). ** Control versus treatment with BMP-2, *p* < 0.05, *n* = 3. (**B**) BMP-2-responsive BRE Luciferase reporter activity in MDA-MB-231 cells transfected with the control or TrkB shRNA. ** Control versus treatment with BMP-2, *p* < 0.05, *n* = 3. (**C**) Western blot analysis of phospho-SMAD1 and SMAD1 expression in the Hs578T, MDA-MB-231 control-shRNA, and TrkB-shRNA cells after stimulation with BMP-2 (5 ng/mL). (**D**) Thymidine incorporation assay showing the proliferation of MDA-MB-231 cells transfected with the control or TrkB shRNA and treated with various BMP-2 concentrations. The points represent the means from three measurements ± SD. ** MDA-MB-231 control-shRNA versus TrkB-shRNA, *p* < 0.05, *n* = 3.

**Figure 3 cancers-12-02095-f003:**
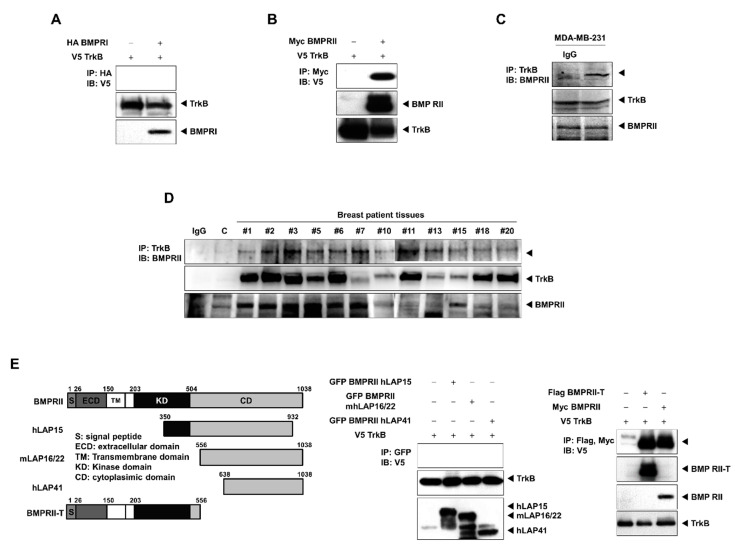
TrkB inhibited BMP signaling following BMP-2 treatment by blocking the interaction between BMP type I receptor (BMPRI) and BMP type II receptor (BMPRII). (**A**) Western blot analysis of whole-cell lysates and hemagglutinin (HA)-tagged immunoprecipitates verifying the presence of V5-TrkB and HA-BMPRI constructs in 293T transfected cells. (**B**) Western blot analysis of whole-cell lysates and Myc-tagged immunoprecipitates verifying the presence of V5-TrkB and Myc-BMPRII constructs in 293T transfected cells. (**C**) The formation of endogenous TrkB-BMPRII complexes in MDA-MB-231 cells, as shown by Western blots of whole-cell lysates and TrkB immunoprecipitates. (**D**) The formation of endogenous TrkB-BMPRII complexes in the tumor tissues of breast cancer patients, as shown by Western blots of whole-cell lysates and TrkB immunoprecipitates. (**E**) The identification of the BMPRII region that interacted with TrkB. Western blot analysis of whole-cell lysates and green fluorescent protein (GFP)-, Flag-, or Myc-tagged immunoprecipitates obtained from 293T cells transfected with V5-TrkB, Myc-BMPRII, or a BMPRII-deleted construct.

**Figure 4 cancers-12-02095-f004:**
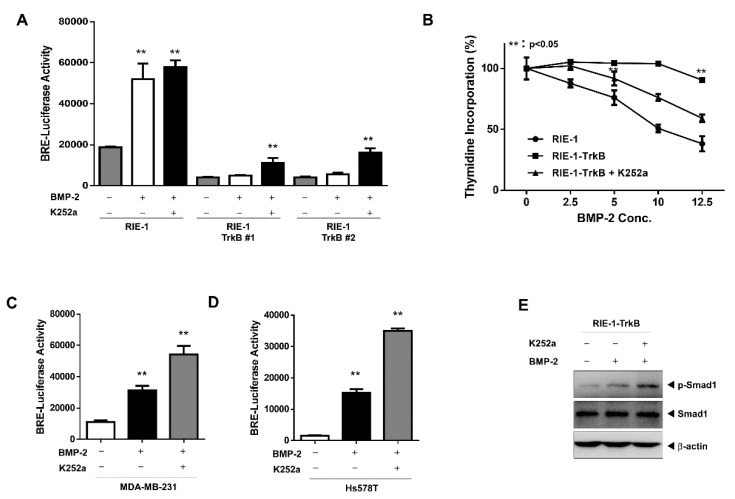
Tyrosine kinase activity of TrkB is required to inhibit BMP signaling. (**A**) BMP-2-responsive BRE Luciferase reporter activity in RIE-1 and RIE-1-TrkB cells following treatment with K252a (50 nM) and BMP-2 (5 ng/mL). Luciferase activity was measured 24 h after treatment with BMP-2 (5 ng/mL). ** Control versus treatment with K252a, *p* < 0.05, *n* = 3. (**B**) Thymidine incorporation assay of RIE-1 and RIE-1-TrkB cells treated with K252a (50 nM) and various BMP-2 concentrations. The points represent the means from three measurements ±SD. (**C**,**D**) BMP-2-responsive BRE Luciferase reporter activity in MDA-MB-231 and Hs578T cells treated with K252a (50 nM). ** Control versus treatment with BMP-2, *p* < 0.05, *n* = 3. (**E**) Western blot analysis of phospho-SMAD1 and SMAD1 expression in RIE-1 and RIE-1-TrkB cells with or without BMP-2 (5 ng/mL) or K252a (50 nM) treatment.

**Figure 5 cancers-12-02095-f005:**
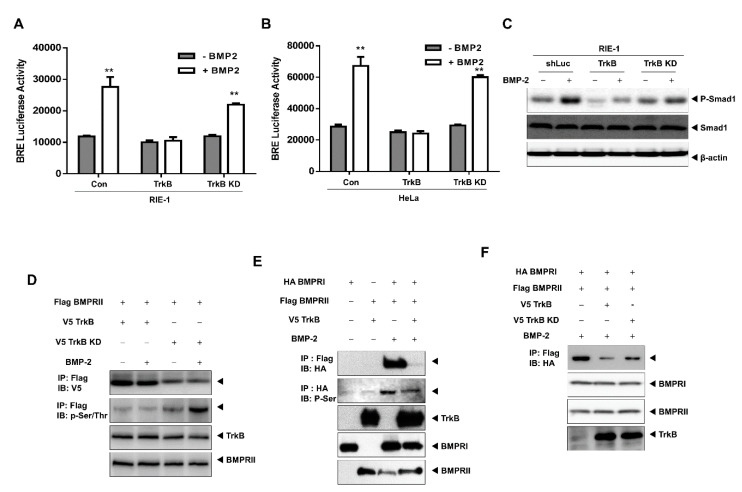
Loss of TrkB kinase activity restored BMP signaling following treatment with BMP-2 by enhancing interaction between BMPRI and BMPRII. (**A**,**B**) BMP-2-responsive BRE Luciferase reporter activity in RIE-1 and HeLa cells transfected with the TrkB or RIE-1-TrkB KD constructs. ** Control versus treatment with BMP-2, *p* < 0.05, *n* = 3. (**C**) Western blot analysis of phospho-SMAD1 and SMAD1 expression in RIE-1, RIE-1-TrkB, and RIE-1-TrkB KD cells with or without BMP-2 (5 ng/mL) treatment. (**D**) Western blot analysis of whole-cell lysates and Flag-tagged immunoprecipitates of 293T cells transfected with the V5-TrkB, V5-TrkB KD, or Flag-BMPRII constructs and with or without BMP-2 (5 ng/mL) treatment. (**E**) Western blot analysis of whole-cell lysates and Flag-tagged immunoprecipitates of 293T cells transfected with the HA-BMPRI, Flag-BMPRII, or V5-TrkB constructs and with or without BMP-2 (5 ng/mL) treatment. (**F**) Western blot analysis of whole-cell lysates and Flag-tagged immunoprecipitates of 293T cells transfected with the HA-BMPRI, Flag-BMPRII, V5-TrkB, or V5-TrkB KD constructs and treated with/without BMP-2 (5 ng/mL).

**Figure 6 cancers-12-02095-f006:**
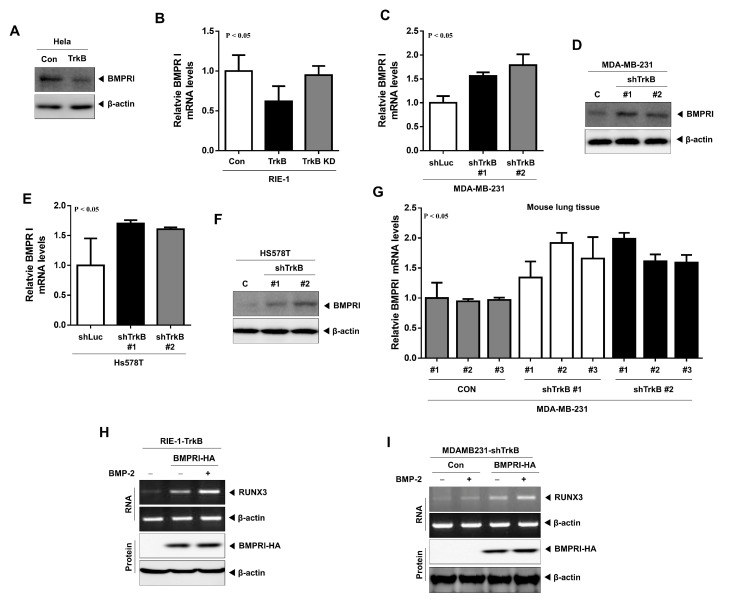
TrkB suppressed BMP-mediated RUNX3 expression by depleting BMPRI. (**A**) Western blot analysis of BMPRI expression in HeLa and HeLa-TrkB cells. Loading control: β-actin. (**B**) Relative expression of the BMPRI mRNA in RIE-1, RIE-1-TrkB, and RIE-TrkB KD cells, as determined by quantitative reverse transcription polymerase chain reaction (RT-PCR). Loading control: 18S. *p* < 0.05, *t*-test. (**C**) Relative expression of the BMPRI mRNA in MDA-MB-231 cells transfected with the control or TrkB shRNA, as determined by quantitative RT-PCR. Loading control: 18S. *p* < 0.05, *t*-test. (**D**) Western blot analysis of BMPRI expression in MDA-MB-231 cells transfected with the control or TrkB shRNA. Loading control: β-actin. (**E**) Relative expression of the BMPRI mRNA in Hs578T cells transfected with the control or TrkB shRNA, as determined by quantitative RT-PCR. Loading control: 18S. *p* < 0.05, *t*-test. (**F**) Western blot analysis of BMPRI expression in Hs578T cells transfected with the control or TrkB shRNA. Loading control: β-actin. (**G**) Relative expression of the BMPRI mRNA in the lungs of mice injected with either MDA-MB-231 control-shRNA or TrkB-shRNA cells, as determined by quantitative RT-PCR. Loading control: 18S. *p* < 0.05, *t*-test. (**H**) Relative expression of the RUNX3 mRNA and protein in RIE-TrkB cells transfected with BMPRI and treated with BMP-2 (5 ng/mL). (**I**) Relative expression of the RUNX3 mRNA and protein in MDA-MB-231 TrkB-shRNA cells transfected with BMPRI and treated with BMP-2 (5 ng/mL).

**Figure 7 cancers-12-02095-f007:**
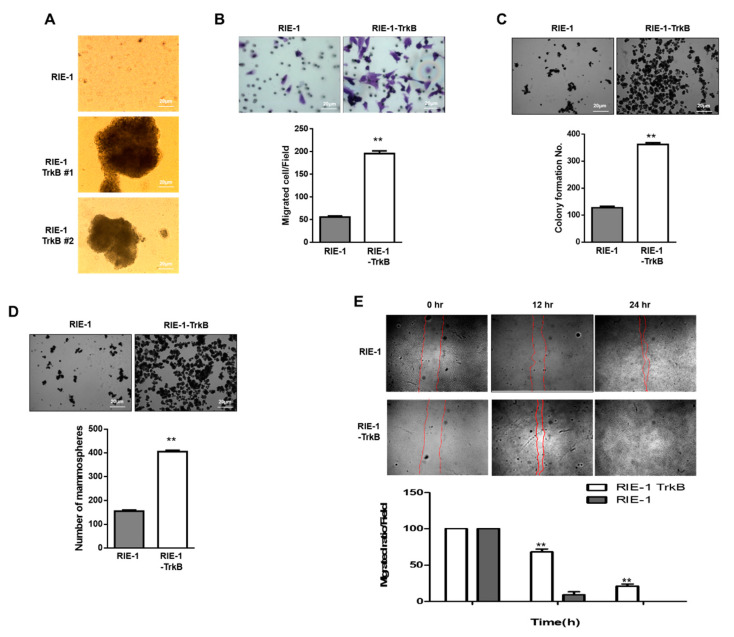
TrkB induced tumor cell migration, survival, and transformation. (**A**) Images of the spheroid colonies formed by RIE-1 or RIE-1-TrkB cells (×200 magnification). Scale bar represents 20 μm. (**B**) Migration assay: bright-field microscopy images and quantification of RIE-1 or RIE-1-TrkB cells (*n* = 3). *p* < 0.0001, *t*-test. Scale bar represents 20 μm. (**C**) Colony-formation assay: bright-field microscopy images and quantification of RIE-1 or RIE-1-TrkB cells (*n* = 3). *p* < 0.0001, *t*-test. The scale bar represents 20 μm. (**D**) Mammosphere formation assay: bright-field microscopy images and quantification of RIE-1 or RIE-1-TrkB cells (*n* = 3). *p* < 0.0001, *t*-test. Scale bar represents 20 μm. (**E**) Wound-healing assay: bright-field microscopy images and quantification of RIE-1 or RIE-1-TrkB cells. ** *p* < 0.05. *n* = 3.

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
