# Peer review of "TrkB Inhibits the BMP Signaling-Mediated Growth Inhibition of Cancer Cells"

_cancers, 2020, doi:10.3390/cancers12082095_

Round 1
Reviewer 1 Report
The authors responded properly to my questions, performed additional experiments, and made intensive changes to improve the quality of the manuscript. In general, the revised version looks much better.
In my opinion, the title of the manuscript still requires some minor changes/modifications. Since the authors did not perform the in vivo studies, it is not correct to make a point about tumor suppression. I suggest to highlight the point about the properties of tumor cells but not about the tumors.
Another concern is about the language quality. Despite the authors corrected the manuscript through English proofreading & editing service with native English speakers, it still requires the language editing.
Just a few examples highlighted in bold font:
1) TrkB removes the BMP-mediated inhibition of small mothers against decapentaplegic 1 (SMAD1) by promoting the formation of the TrkB/BMP type II receptor complex and suppresses RUNX3 by depleting BMP receptor I (BMPRI) expression (lines 18-20).
2)TrkB expression drives the inhibition of an established BMP signaling cascade (line 73)
3) TrkB promotes the ability of tumorigenicity and Metastasis of RIE-1 Cells (line 268)
Author Response
The authors responded properly to my questions, performed additional experiments, and made intensive changes to improve the quality of the manuscript. In general, the revised version looks much better.
In my opinion, the title of the manuscript still requires some minor changes/modifications. Since the authors did not perform the in vivo studies, it is not correct to make a point about tumor suppression. I suggest to highlight the point about the properties of tumor cells but not about the tumors.
Thank you for your suggestion. As your suggestion, we change the title as TrkB inhibits BMP signaling-mediated growth inhibition of cancer cells.
Also, As your suggestion, we modified our manuscript to highlight the point about the properties of tumor cells.
Please see the main text of the manuscript.
Another concern is about the language quality. Despite the authors corrected the manuscript through English proofreading & editing service with native English speakers, it still requires the language editing.
Just a few examples highlighted in bold font:
1) TrkB removes the BMP-mediated inhibition of small mothers against decapentaplegic 1 (SMAD1) by promoting the formation of the TrkB/BMP type II receptor complex and suppresses RUNX3 by depleting BMP receptor I (BMPRI) expression (lines 18-20).
2)TrkB expression drives the inhibition of an established BMP signaling cascade (line 73)
3) TrkB promotes the ability of tumorigenicity and Metastasis of RIE-1 Cells (line 268)
As your suggestion, we corrected and modified the manuscript with a native English speaker.
Reviewer 2 Report
The revised manuscript has a big improvement. However, I have a great concern about the authenticity of the manuscript.
The reason I raised my concerns because Fig 6 and Fig 7 are same figures. This could be a mistake made during revising a manuscript or figures were purposely and intentionally done by authors.
Scale bars were included in the figure but it did not tell us how big the scale bar is?
Please explain.
Author Response
Scale bars were included in the figure but it did not tell us how big the scale bar is?
Please explain.
As your suggestion, we added the size of the scale bar in figure 7 and its legend.